# Airborne Radar STAP Method Based on Deep Unfolding and Convolutional Neural Networks

Bo Zou [1], Weike Feng [1,*] and Hangui Zhu [2]

1 Early Warning and Detection Department, Air Force Engineering University, Xi'an 710051, China; zoubo1999@126.com
2 Electronic Information School, Wuhan University, Wuhan 430072, China; zhg598@hotmail.com
* Correspondence: fengweike007@163.com; Tel.: +86-180-6688-3988

**Abstract:** The lack of independent and identically distributed (IID) training range cells is one of the key factors that limit the performance of conventional space-time adaptive processing (STAP) methods for airborne radar. Sparse recovery (SR)-based and convolutional neural network (CNN)-based STAP methods can obtain high-resolution estimations of the clutter space-time spectrum by using few IID training range cells, so as to realize the clutter suppression effectively. However, the performance of SR-STAP methods usually depends on the SR algorithms, having the problems of parameter setting difficulty, high computational complexity and low accuracy, and the CNN-STAP methods have a high requirement for the nonlinear mapping capability of CNN. To solve these problems, CNNs can be used to reduce the requirements of SR algorithms for parameter setting and iterations, increasing its accuracy, and the clutter space-time spectrum obtained by SR can be used to reduce the network scale of the CNN, resulting in the method proposed in this paper. Based on the idea of deep unfolding (DU), the SR algorithm is unfolded into a deep neural network, whose optimal parameters are obtained by training to improve its convergence performance. On this basis, the SR network and CNN are trained end-to-end to estimate the clutter space-time spectrum efficiently and accurately. The simulation and experimental results show that, compared to the SR-STAP and CNN-STAP methods, the proposed method can improve the clutter suppression performance and have a lower computational complexity.

**Keywords:** space-time adaptive processing (STAP); sparse recovery (SR); convolutional neural network (CNN); deep unfolding (DU)

## 1. Introduction

Airborne radar usually faces complicated ground and/or sea clutter when detecting low-altitude moving targets. The clutter presents space-time coupling characteristics, and its power spectrum broadens severely. In general, one-dimensional methods based on Doppler filtering or spatial beamforming cannot achieve effective target detection. The space-time adaptive processing (STAP) method combines spatial and temporal two-dimensional information to suppress clutter in the space-time domain adaptively, improving the detection performance of moving targets for airborne radar [1,2]. However, conventional STAP methods need to use a certain number of independent and identically distributed (IID) training range cells to estimate the clutter plus noise covariance matrix (CNCM) of the range cell under test (RUT). It has been shown that the number of IID training range cells required by conventional STAP methods is at least twice the system's degree of freedom to ensure that the loss of the output signal-to-clutter-plus-noise ratio (SCNR) is less than 3 dB compared to that of the optimal STAP [3]. In practice, non-ideal factors, such as non-stationary environment, non-homogeneous clutter characteristics and complicated platform motion, make this condition difficult to meet [4–6].

To reduce the requirement for IID training range cells, reduced-dimension STAP methods, reduced-rank STAP methods, direct data domain STAP methods and SR-based

STAP methods have been proposed [7–10]. Among these methods, SR-STAP methods based on the focal under-determined system solver (FOCUSS), alternating direction method of multipliers (ADMM) and sparse Bayesian learning (SBL) algorithm [11–15] can achieve high-resolution estimation of the clutter space-time amplitude spectrum by using a small number of IID training range cells. Then, CNCM can be calculated to suppress clutter and detect targets. However, the SR algorithms used in SR-STAP methods usually have the problems of parameter setting difficulty, high computational complexity and low accuracy. In addition, under the non-side-looking conditions, the clutter sparsity deteriorates, and the performance of SR-STAP methods degrades severely [16].

Recently, based on the idea of image super-resolution reconstruction via deep learning [17,18], a new STAP method based on convolutional neural networks (CNNs) was proposed [19,20]. The CNN-STAP method trains a CNN offline by constructing the dataset that simulates the real clutter environment so that it can reconstruct a high-resolution image from its low-resolution version. Then, the trained CNN is used online to process the low-resolution clutter space-time power spectrum obtained based on a small number of IID training range cells to obtain the high-resolution estimation of the clutter space-time power spectrum, thus constructing the space-time filter for clutter suppression and target detection. CNN-STAP can obtain a high clutter suppression performance, and its online computing complexity is greatly reduced compared to the SR-STAP method. However, due to the small network scale and the relatively poor reconstruction capacity of the CNN constructed in [19], the clutter suppression performance of the CNN-STAP method needs to be further improved. Increasing the network scale can improve the reconstruction capacity of the CNN and the performance of CNN-STAP, but it inevitably increases the computational complexity.

To reduce the requirement for the network reconstruction capacity of the CNN-STAP method, the clutter space-time spectrum obtained by the SR algorithm can be used as the input of the CNN, i.e., the SR-STAP method and the CNN-STAP method can be cascaded. In such a case, due to the high quality of the input clutter space-time spectrum, a high-resolution clutter spectrum can be reconstructed using a small-scaled CNN. Moreover, this method can reduce the requirements for parameter setting and iterations of the SR-STAP method and can improve its estimation accuracy. Based on reasonable iteration parameters and a small number of iterations, the SR algorithm can be used to process a small amount of IID training range cell data to obtain the high-resolution clutter space-time spectrum with some errors. Then, the CNN can be used to improve the estimation accuracy. Furthermore, based on the idea of deep unfolding (DU) [21–25], the SR algorithm can be unfolded to form a deep neural network, and its optimal parameters under a certain number of iterations can be obtained via training, which can help to improve the accuracy of clutter space-time spectrum estimation. Then, the CNN can be used to obtain the final clutter space-time spectrum estimation result.

Based on the ideas mentioned above, a DU-CNN-STAP method is proposed in this paper. First, the airborne radar signal model is established, and the SR-STAP and CNN-STAP methods are briefly introduced. Then, the processing framework, network structure, dataset construction and training methods of the DU-CNN-STAP method are introduced in detail. Finally, the performance and advantages of the DU-CNN-STAP method are verified via simulation and experimental data. The results show that, compared to the SR-STAP and CNN-STAP methods, the proposed method can improve the clutter suppression performance and have a lower computational complexity.

## 2. Signal Model and STAP Methods

### 2.1. Signal Model

As shown in Figure 1, an airborne radar with a uniform linear array (ULA) moves along the $y$-axis at an altitude of $H$ with a constant speed of $v$. The number of elements in the ULA is $M$, and the spacing of adjacent elements is $d$. The angle between the ULA and

the radar moving direction is $\theta_e$. The radar transmits and receives $N$ pulses in a coherent processing interval (CPI) with a pulse repetition interval of $T_r$.

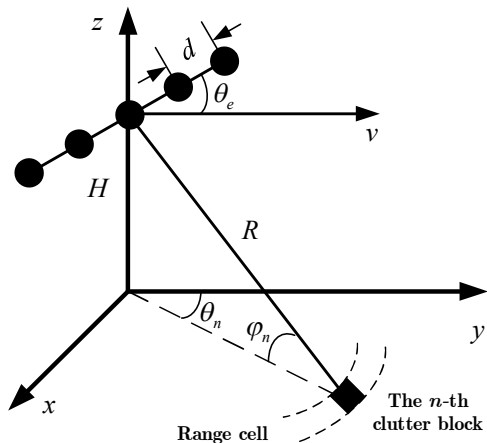

**Figure 1.** Airborne radar target detection model.

Without considering range ambiguity, the clutter range cell on the ground/sea surface is supposed to consist of $N_c$ clutter blocks, and their scattering coefficients are assumed to be mutually independent. Thus, the clutter-plus-noise component of the radar space-time echoed signal $\mathbf{x}$ can be given as

$$\mathbf{x}_c + \mathbf{x}_n = \sum_{n=1}^{N_c} \sigma_{c;n} \mathbf{v}(f_{c;d,n}, f_{c;s,n}) + \mathbf{x}_n = \sum_{n=1}^{N_c} \sigma_{c;n}(\mathbf{v}_d(f_{c;d,n}) \otimes \mathbf{v}_s(f_{c;s,n})) + \mathbf{x}_n \tag{1}$$

where $\mathbf{x}_n$ is the complex Gaussian white noise signal with a mean of 0 and variance of $\sigma_n^2$, $\otimes$ denotes the Kronecker product, $\sigma_{c;n}$ denotes the scattering coefficient of the $n-$th ($n = 1, 2, \ldots, N_c$) clutter block, $\mathbf{v}_d(f_{c;d,n})$ and $\mathbf{v}_s(f_{c;s,n})$ denote the temporal and spatial steering vectors of the $n-$th clutter block, expressed as

$$\begin{cases} \mathbf{v}_d(f_{c;d,n}) = [1, \exp(j2\pi f_{c;d,n}), \cdots, \exp(j2\pi(N-1)f_{c;d,n})]^T \in \mathbb{C}^{N \times 1} \\ \mathbf{v}_s(f_{c;s,n}) = [1, \exp(j2\pi f_{c;s,n}), \cdots, \exp(j2\pi(M-1)f_{c;s,n})]^T \in \mathbb{C}^{M \times 1} \end{cases} \tag{2}$$

where $(\cdot)^T$ denotes the transpose, $f_{c;d,n}$ and $f_{c;s,n}$ are the normalized Doppler frequency and spatial frequency of the $n-$th clutter block, given by

$$\begin{cases} f_{c;d,n} = \frac{2vT_r}{\lambda} \cos\theta_n \cos\varphi_n \\ f_{c;s,n} = \frac{d}{\lambda} \cos(\theta_n + \theta_e) \cos\varphi_n \end{cases} \tag{3}$$

where $\varphi_n$ and $\theta_n$ denote the elevation angle and azimuth angle of the $n-$th clutter block, respectively, and $\lambda$ denotes the signal wavelength.

According to (1), assuming that clutter and noise are mutually independent, the CNCM can be expressed as

$$\begin{aligned} \mathbf{R}_I &= \mathbf{R}_c + \mathbf{R}_n = E(\mathbf{x}_c \mathbf{x}_c^H) + E(\mathbf{x}_n \mathbf{x}_n^H) \\ &= \sum_{n=1}^{N_c} E(|\sigma_{c;n}|_2) \mathbf{v}(f_{c;d,n}, f_{c;s,n}) \mathbf{v}^H(f_{c;d,n}, f_{c;s,n}) + \sigma_n^2 \mathbf{I}_{NM} \end{aligned} \tag{4}$$

where $E(\cdot)$ denotes the expectation, $(\cdot)^H$ denotes the conjugate transpose, and $\mathbf{I}_{NM}$ denotes the unit matrix with a dimension of $NM \times NM$.

The output of the STAP filter is the inner product of the space-time weighting vector **w** and the radar space-time echoed signal **x**, expressed as

$$\mathbf{y} = \mathbf{w}^{\mathrm{H}}\mathbf{x} \qquad (5)$$

To maintain the target power while minimizing the power of clutter and noise after filtering, the optimal weighting vector of the STAP filter can be calculated as follows:

$$\mathbf{w}_{\mathrm{opt}} = \mathbf{R}_{\mathrm{I}}^{-1}\mathbf{v}_{\mathrm{t}}/(\mathbf{v}_{\mathrm{t}}^{\mathrm{H}}\mathbf{R}_{\mathrm{I}}^{-1}\mathbf{v}_{\mathrm{t}}) \in \mathbb{C}^{NM\times 1} \qquad (6)$$

where $(\cdot)^{-1}$ denotes matrix inversion, and $\mathbf{v}_{\mathrm{t}}$ denotes the space-time steering vector of the target.

In general, a certain number of training range cells without including the target are needed to estimate the CNCM of the RUT. Under the condition that the training range cells and the RUT are IID, via the sample matrix inversion (SMI) method [2], the CNCM of the RUT can be estimated as

$$\hat{\mathbf{R}}_{\mathrm{I}} = \frac{1}{L}\sum_{l=1}^{L}\mathbf{x}_{l}\mathbf{x}_{l}^{\mathrm{H}} \qquad (7)$$

where $L$ is the number of IID training range cells, and $\mathbf{x}_{l}$ denotes the space-time echoed signal of the $l-$th training range cell.

According to the RMB criterion [3], the output SCNR loss of the SMI method with respect to the optimal STAP method can be expressed as

$$\mathrm{SCNR}_{\mathrm{loss}} = (L - O + 2)/(L + 1) \qquad (8)$$

where $O = MN$ denotes the system's degree of freedom (DOF).

Equation (8) demonstrates that, to ensure that the output SCNR loss is less than 3 dB, the number of IID training range cells required by the SMI method is at least about twice the system's DOF, i.e., $L$ should be larger than $2O - 3$, which is difficult to meet in a practical non-homogeneous and non-stationary clutter environment.

### 2.2. SR-STAP Method

It can be seen from (1) that the clutter signal is superimposed by the space-time signals corresponding to different clutter blocks. Thus, the entire spatial frequency–Doppler frequency domain can be discretized into $N_{\mathrm{s}} \times N_{\mathrm{d}}$ grids, where $N_{\mathrm{s}} = \rho_{\mathrm{s}}M$, $N_{\mathrm{d}} = \rho_{\mathrm{d}}N$, and $N_{\mathrm{s}}N_{\mathrm{d}} \gg NM$, to approximate the clutter component as

$$\mathbf{x}_{\mathrm{c}} = \sum_{i=1}^{N_{\mathrm{d}}}\sum_{j=1}^{N_{\mathrm{s}}}\gamma_{i,j}\mathbf{v}(f_{\mathrm{d},i}, f_{\mathrm{s},j}) = \mathbf{\Phi}\boldsymbol{\gamma} \qquad (9)$$

where $f_{\mathrm{d},i}$ is the $i-$th Doppler frequency ($i = 1, 2, \ldots, N_d$), $f_{\mathrm{s},j}$ is the $j-$th spatial frequency ($j = 1, 2, \ldots, N_s$), $\mathbf{v}(f_{\mathrm{d},i}, f_{\mathrm{s},j})$ is the space-time steering vector of the $i - j-$th grid, $\gamma_{i,j}$ denotes the complex amplitude of the $i - j-$th grid, $\boldsymbol{\gamma} = [\gamma_{1,1}, \gamma_{2,1}, \cdots, \gamma_{N_{\mathrm{d}},N_{\mathrm{s}}}] \in \mathbb{C}^{N_s N_d \times 1}$ denotes the complex amplitude vector corresponding to all grids, i.e., the clutter space-time amplitude spectrum, and $\mathbf{\Phi}$ denotes the dictionary of space-time steering vectors, given by

$$\mathbf{\Phi} = [\mathbf{v}(f_{\mathrm{d},1}, f_{\mathrm{s},1}), \mathbf{v}(f_{\mathrm{d},2}, f_{\mathrm{s},1}), \cdots, \mathbf{v}(f_{\mathrm{d},N_{\mathrm{d}}}, f_{\mathrm{s},N_{\mathrm{s}}})] \in \mathbb{C}^{NM\times N_s N_d} \qquad (10)$$

Thus, the space-time echoed signal of the $l-$th training range cell without the target can be expressed as

$$\mathbf{x}_{l} = \mathbf{x}_{\mathrm{c},l} + \mathbf{x}_{\mathrm{n},l} = \mathbf{\Phi}\boldsymbol{\gamma}_{l} + \mathbf{x}_{\mathrm{n},l} \qquad (11)$$

Due to the space-time coupling property of clutter, its space-time amplitude spectrum is usually sparse. Based on this property, the SR-STAP method can estimate the clutter space-

time amplitude spectrum by solving a constrained optimization problem, expressed as

$$\hat{\boldsymbol{\gamma}}_l = \arg \min_{\boldsymbol{\gamma}_l} \|\boldsymbol{\gamma}_l\|_0, \quad s.t. \; \|\mathbf{x}_l - \boldsymbol{\Phi}\boldsymbol{\gamma}_l\|_2 \leq \varepsilon \tag{12}$$

where $\| \cdot \|_0$ denotes the $L_0$ norm, $\| \cdot \|_2$ denote the $L_2$ norm, and $\varepsilon$ denotes the noise level.

With $L$ training range cells, (12) can be upgraded to the multiple measurement vectors (MMV) model [14], expressed as

$$\hat{\boldsymbol{\Gamma}} = \arg \min_{\boldsymbol{\Gamma}} \|\boldsymbol{\Gamma}\|_{2,0}, \quad s.t. \; \|\mathbf{X} - \boldsymbol{\Phi}\boldsymbol{\Gamma}\|_F \leq \varepsilon \tag{13}$$

where $\mathbf{X} = [\mathbf{x}_1, \mathbf{x}_2, \ldots, \mathbf{x}_L] \in \mathbb{C}^{NM \times L}$, $\boldsymbol{\Gamma} = [\boldsymbol{\gamma}_1, \boldsymbol{\gamma}_2, \ldots, \boldsymbol{\gamma}_L] \in \mathbb{C}^{N_s N_d \times L}$, $\| \cdot \|_{2,0}$ denotes the $L_0$ norm of the column vector obtained by the $L_2$ norm of each row of a matrix, and $\| \cdot \|_F$ denotes the Frobenius norm.

By solving (12) or (13) with the SR algorithm, the CNCM can be estimated as

$$\hat{\mathbf{R}}_I = \frac{1}{L} \sum_{l=1}^{L} \sum_{i=1}^{N_d} \sum_{j=1}^{N_s} \left| \gamma_{l,i,j} \right|^2 \mathbf{v}(f_{d,i}, f_{s,j}) \mathbf{v}^H(f_{d,i}, f_{s,j}) + \sigma_n^2 \mathbf{I}_{NM} \tag{14}$$

where $\gamma_{l,i,j}$ denotes the complex amplitude of the $i-j-$th grid of the $l-$th training range cell.

Based on the estimated CNCM, according to (6), the weighting vector can be calculated as

$$\hat{\mathbf{w}}_{\text{opt}} = \hat{\mathbf{R}}_I^{-1} \mathbf{v}_t / \left( \mathbf{v}_t^H \hat{\mathbf{R}}_I^{-1} \mathbf{v}_t \right) \in \mathbb{C}^{NM \times 1} \tag{15}$$

The SR-STAP methods can estimate the CNCM accurately by using far fewer IID training range cells than the system's DOF, i.e., $L \ll O$. However, given the clutter space-time amplitude spectrum estimation model shown in (12) and (13), the performance of the SR-STAP methods is usually dependent on the employed SR algorithm. Although a series of SR algorithms have been proposed, there are still some problems, such as parameter setting difficulty, high computing complexity and low accuracy. In addition, under the non-side-looking conditions, the clutter sparsity deteriorates, and thus the performance of the SR-STAP methods degrades severely.

### 2.3. CNN-STAP Method

Based on the idea of deep-learning-based image super-resolution reconstruction, the CNN-STAP method [19,20] estimates a low-resolution clutter space-time power spectrum first based on a small number of IID training range cells. Then, the CNN is used to reconstruct a high-resolution clutter space-time power spectrum. Finally, the CNCM and the space-time adaptive weighting vector are calculated for clutter suppression and target detection. The specific steps can be summarized as follows.

Assuming there are $L$ training range cells ($L \ll O$) and the corresponding space-time echoed signal matrix is $\mathbf{X}$, the low-resolution clutter space-time power spectrum $\mathbf{Y} \in \mathbb{C}^{N_d \times N_s}$ can be obtained via the Fourier-transform-based digital beam forming (DBF) algorithm, acting as the input data of the CNN, expressed as

$$\mathbf{Y}(i,j) = \left\| \mathbf{v}(f_{d,i}, f_{s,j})^H \mathbf{X} \right\|_2^2 / L \tag{16}$$

Then, based on the theoretical clutter covariance matrix $\mathbf{R}_c$, the minimum variance distortion-less response (MVDR) spectrum estimation method [26] is used to obtain a high-resolution clutter space-time power spectrum $\mathbf{Z}_T \in \mathbb{C}^{N_d \times N_s}$, acting as the output label data of CNN, expressed as

$$\mathbf{Z}_T(i,j) = \frac{1}{\left| \mathbf{v}(f_{d,i}, f_{s,j})^H \mathbf{R}_c^{-1} \mathbf{v}(f_{d,i}, f_{s,j}) \right|} \tag{17}$$

The process of reconstructing a high-resolution clutter space-time power spectrum from the low-resolution spectrum $\mathbf{Y}$ via a CNN can be expressed as [19]

$$\hat{\mathbf{Z}}_{\mathrm{C}} = \mathcal{F}_{\mathrm{CNN}}(\mathbf{Y}, \boldsymbol{\Theta}_{\mathrm{C}}) \tag{18}$$

where $\mathcal{F}_{\mathrm{CNN}}(\cdot)$ denotes a nonlinear transform operation conducted on the clutter space-time power spectrum, acquiring a high-resolution spectrum from its low-resolution version, $\boldsymbol{\Theta}_{\mathrm{C}} = \{\mathbf{W}_1, \mathbf{W}_2, \cdots, \mathbf{W}_E, \mathbf{b}_1, \mathbf{b}_2, \cdots, \mathbf{b}_E\}$ denotes the network parameters of the CNN, $\mathbf{W}_e$ ($e = 1, 2, \ldots, E$) denotes the convolutional kernel with a dimension of $c_e \times f_e \times f_e \times n_e$, and $\mathbf{b}_e$ denotes the bias vector with a dimension of $n_e$.

By constructing a dataset to train the CNN, the optimal CNN parameters can be obtained as

$$\boldsymbol{\Theta}_{\mathrm{C}}^* = \arg\min_{\boldsymbol{\Theta}_{\mathrm{C}}} \frac{1}{P} \sum_{p=1}^{P} \left\| \mathcal{F}_{\mathrm{CNN}}(\mathbf{Y}_p, \boldsymbol{\Theta}_{\mathrm{C}}) - \mathbf{Z}_{\mathrm{T},p} \right\|_{\mathrm{F}}^2 \tag{19}$$

where $\mathbf{Y}_p$ denotes the $p-$th low-resolution clutter space-time power spectrum input, $\mathbf{Z}_{\mathrm{T},p}$ denotes the $p-$th high-resolution clutter space-time power spectrum label, $\{\mathbf{Y}_p\}_{p=1}^{P}$ and $\{\mathbf{Z}_{\mathrm{T},p}\}_{p=1}^{P}$ form the training dataset for the CNN, and $P$ is the dataset size.

Finally, the actual space-time echoed signal matrix can be processed online by the trained CNN. The low-resolution clutter space-time power spectrum $\mathbf{Y}$ can be transformed to obtain the high-resolution clutter space-time power spectrum $\hat{\mathbf{Z}}_{\mathrm{C}} = \mathcal{F}_{\mathrm{CNN}}(\mathbf{Y}, \boldsymbol{\Theta}_{\mathrm{C}}^*)$, and thus, the CNCM can be estimated as [19]

$$\hat{\mathbf{R}}_{\mathrm{I}} = \sum_{i=1}^{N_{\mathrm{d}}} \sum_{j=1}^{N_{\mathrm{s}}} \hat{\mathbf{Z}}_{\mathrm{C}}(i,j) \mathbf{v}(f_{\mathrm{d},i}, f_{\mathrm{s},j}) \mathbf{v}^{\mathrm{H}}(f_{\mathrm{d},i}, f_{\mathrm{s},j}) + \sigma_{\mathrm{n}}^2 \mathbf{I}_{NM} \tag{20}$$

Similar to the SR-STAP method, the CNN-STAP method can estimate $\hat{\mathbf{R}}_{\mathrm{I}}$ accurately with a small number of IID training range cells. However, due to the poor quality (e.g., low resolution and high sidelobe level) of the clutter space-time power spectrum generated by the Fourier transform, the CNN-STAP method places high demands on the reconstruction capability of CNN. Increasing the network scale can improve the reconstruction capability of the CNN and the performance of CNN-STAP, but the increase of computing complexity is inevitable.

## 3. DU-CNN-STAP

To solve the problems of SR-STAP and CNN-STAP simultaneously, the DU-CNN-STAP (deep unfolding and CNN-based STAP) method is proposed in this paper, as shown in Figure 2. The specific operations are summarized as follows.

Step 1. The space-time echoed signal $\mathbf{x}$ is input into the DU-CNN network (as indicated by the dashed box in Figure 2). The high-resolution clutter space-time power spectrum estimation $\hat{\mathbf{Z}}_{\mathrm{DC}} \in \mathbb{C}^{N_{\mathrm{d}} \times N_{\mathrm{s}}}$ is obtained via the nonlinear transform of DU-CNN.

Step 2. By replacing $\hat{\mathbf{Z}}_{\mathrm{C}}$ in (20) with $\hat{\mathbf{Z}}_{\mathrm{DC}}$, the CNCM $\hat{\mathbf{R}}_{\mathrm{I}}$ is estimated.

Step 3. The space-time adaptive weighting vector $\hat{\mathbf{w}}_{\mathrm{opt}}$ is estimated according to (15), based on which clutter suppression and target detection are conducted.

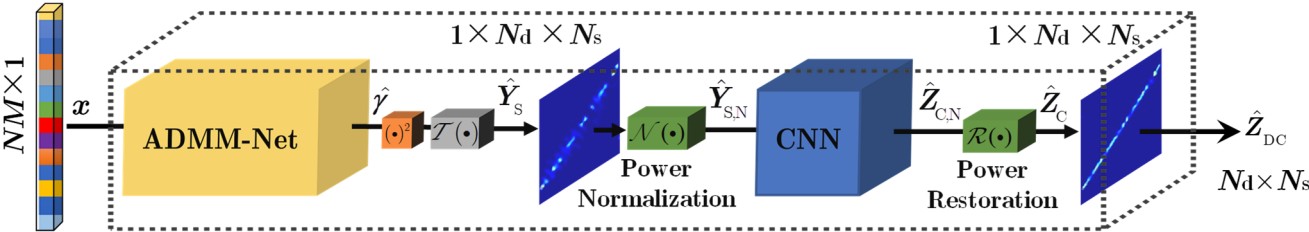

**Figure 2.** Processing framework of DU-CNN-STAP.

DU-CNN-STAP implements a nonlinear transform from the space-time echoed signal **x** directly to the high-resolution clutter power spectrum, i.e., $\hat{\mathbf{Z}}_{\text{DC}} = \mathcal{F}_{\text{DU-CNN}}(\mathbf{x})$. The key of the method is the DU-CNN network. The following points should be noted: (1) The ADMM-Net is a solving network for the SR problem in (12) with the network parameter as $\mathbf{\Theta}_{\text{A}}$, which can help to achieve fast acquisition of the clutter space-time amplitude spectrum $\hat{\boldsymbol{\gamma}} \in \mathbb{C}^{N_{\text{d}} N_{\text{s}} \times 1}$. (2) The squarer module $(\cdot)^2$ completes the conversion from the space-time amplitude spectrum to the space-time power spectrum. The transform $\mathcal{T}(\cdot)$ converts the data dimension from $N_{\text{d}} N_{\text{s}} \times 1$ to $1 \times N_{\text{d}} \times N_{\text{s}}$. The output data can be expressed as $\hat{\mathbf{Y}}_{\text{S}} = \mathcal{T}(|\hat{\boldsymbol{\gamma}}|_2) \in \mathbb{C}^{1 \times N_{\text{d}} \times N_{\text{s}}}$. (3) The power normalization module normalizes the space-time spectrum data to improve the network convergence and obtains the input data of the CNN, expressed as $\hat{\mathbf{Y}}_{\text{S,N}} = \mathcal{N}(\hat{\mathbf{Y}}_{\text{S}}) = \hat{\mathbf{Y}}_{\text{S}}/P_{\text{S}} \in \mathbb{C}^{1 \times N_{\text{d}} \times N_{\text{s}}}$, where $P_{\text{S}} = \left\| \hat{\mathbf{Y}}_{\text{S}} \right\|_{\text{F}}$. (4) The CNN module is the space-time power spectrum reconstruction network with the parameter as $\mathbf{\Theta}_{\text{C}}$, which implements the estimation of the normalized high-resolution clutter space-time power spectrum $\hat{\mathbf{Z}}_{\text{C,N}} \in \mathbb{C}^{1 \times N_{\text{d}} \times N_{\text{s}}}$. (5) The power restoring module performs clutter power restoring on $\hat{\mathbf{Z}}_{\text{C,N}}$, whose output is $\hat{\mathbf{Z}}_{\text{C}} = \mathcal{R}(\hat{\mathbf{Z}}_{\text{C,N}}) = P_{\text{S}} \hat{\mathbf{Z}}_{\text{C,N}} \in \mathbb{C}^{1 \times N_{\text{d}} \times N_{\text{s}}}$. Finally, the network output $\hat{\mathbf{Z}}_{\text{DC}} \in \mathbb{C}^{N_{\text{d}} \times N_{\text{s}}}$ of DU-CNN can be obtained.

With a small number of layers (i.e., iterations), ADMM-Net can obtain a high-resolution estimation of the clutter space-time spectrum, and the CNN can further improve the estimation accuracy. Thus, the DU-CNN-STAP method can effectively solve the problems of parameter setting difficulty, high computational complexity and low accuracy of the SR-STAP method. In addition, unlike the CNN-STAP method, which uses the low-resolution clutter space-time spectrum as the input of the reconstruction network, the input of the reconstruction network in the proposed DU-CNN-STAP method is the high-resolution clutter space-time spectrum. Thus, the DU-CNN-STAP method can reduce the requirements for the nonlinear transform capability, the network scale and the computational complexity of the reconstruction network.

In the presence of range ambiguity, the SR problems shown in (12) and (13) can still be established, and ADMM-Net can still be used [16]. Thus, in the presence of range ambiguity, the proposed DU-CNN-STAP method is still applicable. In the following, the DU-CNN network in the DU-CNN-STAP method is introduced in detail, including its network structure, dataset construction and training method. It should be noted that this paper only considers the case with a single training range cell, i.e., $L = 1$. The processing of multiple training range cells can be implemented via a simple extension of the proposed method. Thus, the subscript $l$ is ignored in the following.

### 3.1. Network Structure

#### 3.1.1. ADMM-Net

Because the $L_0$ norm is a discontinuous function, the complexity of solving (12) is high. Thus, (12) is often solved by transforming it into an $L_1$ convex optimization problem, expressed as

$$\hat{\boldsymbol{\gamma}} = \arg \min_{\boldsymbol{\gamma}} \|\boldsymbol{\gamma}\|_1, \quad s.t. \ \|\mathbf{x} - \mathbf{\Phi}\boldsymbol{\gamma}\|_2 \leq \varepsilon \tag{21}$$

Introducing an auxiliary variable **r**, (21) can be transformed into

$$\{\hat{\boldsymbol{\gamma}}, \hat{\mathbf{r}}\} = \arg \min_{\boldsymbol{\gamma}, \mathbf{r}} \|\boldsymbol{\gamma}\|_1 + \frac{1}{2\rho} \|\mathbf{r}\|_2^2 \quad s.t. \ \mathbf{\Phi}\boldsymbol{\gamma} + \mathbf{r} = \mathbf{x} \tag{22}$$

where $\rho > 0$ denotes the regularization factor.

The augmented Lagrange function of (22) is given by

$$\{\hat{\boldsymbol{\gamma}}, \hat{\mathbf{r}}, \hat{\boldsymbol{\lambda}}\} = \arg \min_{\boldsymbol{\gamma}, \mathbf{r}, \boldsymbol{\lambda}} \|\boldsymbol{\gamma}\|_1 + \frac{1}{2\rho} \|\mathbf{r}\|_2^2 - \langle \boldsymbol{\lambda}, \mathbf{\Phi}\boldsymbol{\gamma} + \mathbf{r} - \mathbf{x} \rangle + \frac{\beta}{2} \|\mathbf{\Phi}\boldsymbol{\gamma} + \mathbf{r} - \mathbf{x}\|_2^2 \tag{23}$$

which can be transformed into $\{\hat{\boldsymbol{\gamma}}, \hat{\mathbf{r}}, \hat{\boldsymbol{\lambda}}\} = \arg\min_{\boldsymbol{\gamma},\mathbf{r},\boldsymbol{\lambda}} \|\boldsymbol{\gamma}\|_1 + \frac{1}{2\rho}\|\mathbf{r}\|_2^2 + \frac{\beta}{2}\left\|\boldsymbol{\Phi}\boldsymbol{\gamma} + \mathbf{r} - \mathbf{x} - \frac{\boldsymbol{\lambda}}{\beta}\right\|_2^2 - \frac{\boldsymbol{\lambda}^2}{2\beta}$ with $\boldsymbol{\lambda} \in \mathbb{C}^{NM\times1}$ as the Lagrange multiplier and $\beta > 0$ as the quadratic penalty factor.

Given the initial values $\left\{\boldsymbol{\gamma}^{(0)}, \mathbf{r}^{(0)}, \boldsymbol{\lambda}^{(0)}\right\}$, the ADMM algorithm solves (23) via the following three steps with $K$ iterations alternately [27]:

$$
\begin{cases}
\mathbf{r}^{(k)} = \arg\min_{\mathbf{r}} \frac{1}{2\rho}\|\mathbf{r}\|_2^2 + \frac{\beta}{2}\left\|\boldsymbol{\Phi}\boldsymbol{\gamma}^{(k-1)} + \mathbf{r} - \mathbf{x} - \frac{\boldsymbol{\lambda}^{(k-1)}}{\beta}\right\|_2^2 \\
\boldsymbol{\gamma}^{(k)} = \arg\min_{\boldsymbol{\gamma}} \|\boldsymbol{\gamma}\|_1 + \frac{\beta}{2}\left\|\boldsymbol{\Phi}\boldsymbol{\gamma} + \mathbf{r}^{(k)} - \mathbf{x} - \frac{\boldsymbol{\lambda}^{(k-1)}}{\beta}\right\|_2^2 \\
\boldsymbol{\lambda}^{(k)} = \boldsymbol{\lambda}^{(k-1)} - \beta\left(\boldsymbol{\Phi}\boldsymbol{\gamma}^{(k)} + \mathbf{r}^{(k)} - \mathbf{x}\right)
\end{cases}
\tag{24}
$$

where $\mathbf{r}^{(k)}$, $\boldsymbol{\gamma}^{(k)}$ and $\boldsymbol{\lambda}^{(k)}$ denote the estimation of $\mathbf{r}$, $\boldsymbol{\gamma}$ and $\boldsymbol{\lambda}$ in the $k-$th $(k = 1, 2, \cdots, K)$ iteration, respectively.

The solutions of the sub-problems in (24) are given by [27]

$$
\begin{cases}
\mathbf{r}^{(k)} = \frac{\rho}{1+\rho\beta}\left(\boldsymbol{\lambda}^{(k-1)} - \beta\boldsymbol{\Phi}\boldsymbol{\gamma}^{(k-1)} + \beta\mathbf{x}\right) \\
\boldsymbol{\gamma}^{(k)} = \mathcal{S}\left(\boldsymbol{\gamma}^{(k-1)} + \frac{\tau}{\rho\beta}\boldsymbol{\Phi}^{\mathrm{H}}\mathbf{r}^{(k)}, \frac{\tau}{\beta}\right) \\
\boldsymbol{\lambda}^{(k)} = \boldsymbol{\lambda}^{(k-1)} - \beta\left(\boldsymbol{\Phi}\boldsymbol{\gamma}^{(k)} + \mathbf{r}^{(k)} - \mathbf{x}\right)
\end{cases}
\tag{25}
$$

where $\mathcal{S}(\cdot)$ denotes the soft threshold operator [28] and $\tau$ denotes the iteration step size.

The ADMM algorithm can obtain a high-resolution estimation $\hat{\boldsymbol{\gamma}} = \boldsymbol{\gamma}^{(K)}$ with $K$ iterations and the iteration parameters as $\rho$, $\beta$ and $\tau$. Then, the CNCM and the adaptive weighting vector can be calculated according to (14) and (15). However, ADMM is a model-driven algorithm, whose parameters need to be given in advance. In practicality, the setting of parameters is generally difficult. Improper parameter settings affect the convergence performance of the ADMM algorithm, resulting in a high computing complexity of solving (12) and a low estimation performance of the clutter space-time amplitude spectrum. Even if the parameters can be set properly, using the same parameters for each iteration does not guarantee a best convergence performance for the ADMM algorithm. To solve this problem, the ADMM algorithm with a specific number of iterations can be unfolded into a deep neural network based on the idea of deep unfolding. The learning approach can then be used to obtain optimal parameters for different iterations, improving the convergence performance of ADMM.

As shown in Figure 3, the ADMM algorithm with $K$ iterations is unfolded into a $K$-layer ADMM-Net, whose inputs are the space-time echoed signal $\mathbf{x} \in \mathbb{C}^{NM\times1}$ and the space-time steering vector dictionary $\boldsymbol{\Phi}$, and the parameters are $\boldsymbol{\Theta}_{\mathrm{A}} = \left\{\boldsymbol{\Theta}_{\mathrm{A}}^{(k)}\right\}_{k=1}^{K} = \{\rho_k, \beta_k, \tau_k\}_{k=1}^{K}$. The output of each layer is the Lagrange multiplier $\boldsymbol{\lambda}^{(k)} \in \mathbb{C}^{NM\times1}$, the auxiliary variable $\mathbf{r}^{(k)} \in \mathbb{C}^{NM\times1}$ and the space-time amplitude spectrum $\boldsymbol{\gamma}^{(k)} \in \mathbb{C}^{N_d N_s \times1}$. The final output of ADMM-Net is $\hat{\boldsymbol{\gamma}} = \boldsymbol{\gamma}^{(K)}$, and the nonlinear function $F_k(\cdot)$ given in (26) is the same as (25).

$$
\left\{\mathbf{r}^{(k)}, \boldsymbol{\gamma}^{(k)}, \boldsymbol{\lambda}^{(k)}\right\} = F_k(\mathbf{x}, \boldsymbol{\Phi}, \boldsymbol{\lambda}^{(k-1)}, \boldsymbol{\gamma}^{(k-1)}, \mathbf{r}^{(k-1)}, \boldsymbol{\Theta}_{\mathrm{A}}^{(k)})
\tag{26}
$$

During the data-driven network training, $3K$ network parameters $\{\rho_k, \beta_k, \tau_k\}_{k=1}^{K}$ of ADMM-Net are adaptively tuned. This allows ADMM-Net to obtain a higher convergence performance than that of the ADMM algorithm with the same number of iterations, thus improving the estimation accuracy of the clutter space-time amplitude spectrum.

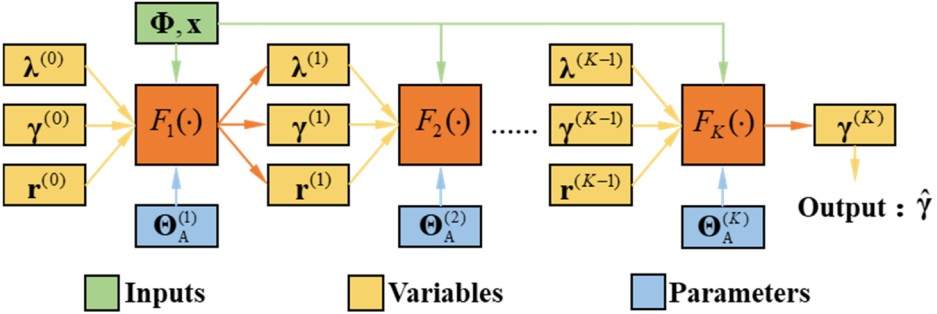

**Figure 3.** Network structure of ADMM-Net.

### 3.1.2. CNN

The space-time power spectrum reconstruction network in DU-CNN is a CNN with $E$ two-dimensional convolutional layers, as shown in Figure 4. The input of the CNN is the normalized clutter space-time power spectrum $\hat{\mathbf{Y}}_{\text{S,N}} = \mathcal{T}(|\hat{\gamma}|_2)/P_{\text{S}}$ obtained by transforming the output $\hat{\gamma}$ of ADMM-Net, whose dimension is $1 \times N_{\text{d}}N_{\text{s}}$, into a number of channels as 1 and a length × width as $N_{\text{d}} \times N_{\text{s}}$. The output of the CNN is the normalized high-resolution clutter space-time power spectrum $\hat{\mathbf{Z}}_{\text{C,N}}$. Because each convolutional layer uses a zero-padding operation, only the number of channels is changed in the processing procedure, and the length × width keeps constant. Thus, the dimension of $\hat{\mathbf{Z}}_{\text{C,N}}$ is also $1 \times N_{\text{d}} \times N_{\text{s}}$.

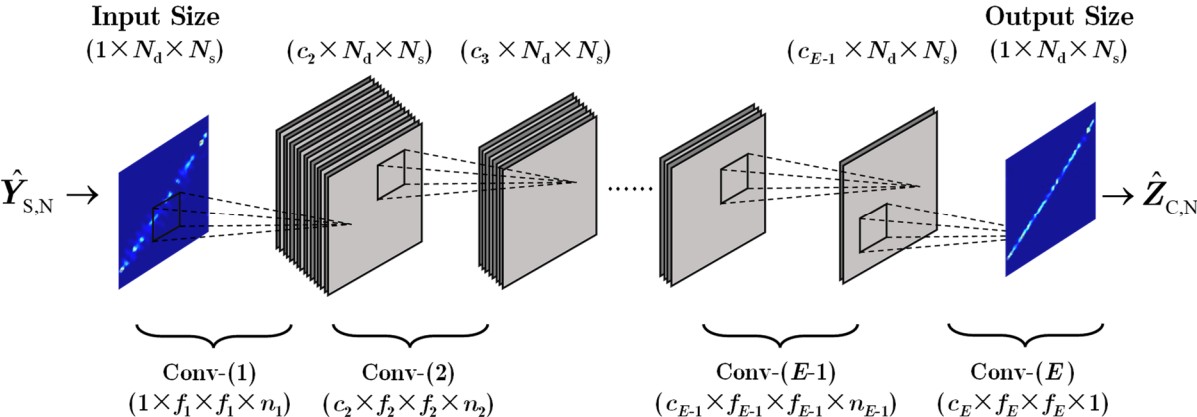

**Figure 4.** Network structure of CNN.

The network parameters of CNN are $\mathbf{\Theta}_{\text{C}} = \{\mathbf{W}_e, \mathbf{b}_e\}_{e=1}^{E}$, where $\mathbf{W}_e$ denotes the convolutional kernel with a dimension of $c_e \times f_e \times f_e \times n_e$, $c_e$ is the number of input channels, $f_e$ is the length and width of the convolutional kernel, $n_e$ is the number of convolutional kernels (i.e., the number of output channels), and $\mathbf{b}_e$ is the bias vector with a dimension of $n_e$. With $*$ denoting the convolutional operation, the operations of each convolutional layer in Figure 4 can be summarized as follows.

(1) Conv-(1) is used to extract features from the input clutter space-time power spectrum, which adopts the ReLU activation function. The specific operation can be expressed as

$$\hat{\mathbf{Z}}_{\text{C,N}}^{(1)} = \max\left(0, \mathbf{W}_1 * \hat{\mathbf{Y}}_{\text{S,N}} + \mathbf{b}_1\right) \tag{27}$$

(2) Conv-(2)~Conv-($E-1$) realize the nonlinear mapping of features, where the ReLU activation function is adopted. The specific operation can be expressed as

$$\hat{\mathbf{Z}}_{\text{C,N}}^{(e)} = \max\left(0, \mathbf{W}_e * \hat{\mathbf{Z}}_{\text{C,N}}^{(e-1)} + \mathbf{b}_e\right), \quad e = 2, 3, \cdots, E-1 \tag{28}$$

(3) Conv-(*E*) is the image reconstruction layer, where the high-resolution clutter space-time power spectrum is output. The specific operation can be expressed as

$$\hat{\mathbf{Z}}_{\text{C,N}} = \mathbf{W}_E * \hat{\mathbf{Z}}_{\text{C,N}}^{(E-1)} + \mathbf{b}_E \tag{29}$$

*3.2. Dataset Construction*

Similar to most deep neural networks, DU-CNN adopts the supervised learning method, i.e., it is trained based on the input data and its corresponding labels. In this paper, the following steps are used to construct a sufficient and complete dataset for DU-CNN to guarantee the clutter space-time spectrum estimation performance and the generalization capability of DU-CNN.

Step 1. The parameters including the airplane height $H$, the number of ULA elements $M$, the number of CPI pulses $N$, the array spacing $d$ and the wavelength $\lambda$ are assumed to be unchanged. The non-side-looking angle, the elevation angle, the clutter ridge slope and the clutter-to-noise ratio (CNR) corresponding to different range cells all obey the uniform random distribution, i.e., $\theta_e \sim U(\theta_{\min}, \theta_{\max})$, $\varphi \sim U(\varphi_{\min}, \varphi_{\max})$, $\beta = 2vT_r/\lambda \sim U(\beta_{\min}, \beta_{\max})$, and $\text{CNR} \sim U(\text{CNR}_{\min}, \text{CNR}_{\max})$. According to (1), $P$ radar space-time echoed signals $\{\mathbf{x}_p\}_{p=1}^P$ are generated as the input dataset for DU-CNN, with each range cell containing a total of $N_c$ clutter blocks. The clutter blocks distribute in the azimuth range $[0, \pi]$ uniformly and have scattering coefficients that obey a complex Gaussian distribution, whose power is determined by the CNR.

Step 2. The spatial frequency $[f_{s,\min}, f_{s,\max}]$ and Doppler frequency $[f_{d,\min}, f_{d,\max}]$ are discretized into $N_s = \rho_s M$ and $N_d = \rho_d N$ grids, respectively. The dictionary of space-time steering vectors $\boldsymbol{\Phi}$ is constructed according to (10). Different combinations of iteration parameters are set according to the convergence conditions of the ADMM algorithm. Based on (12), all the space-time echoed signals $\{\mathbf{x}_p\}_{p=1}^P$ are processed by ADMM. The combination of $\rho = \rho_0$, $\beta = \beta_0$, $\tau = \tau_0$ and $K = K_0$ with the best estimation performance is obtained as the final parameters of ADMM. The estimated clutter space-time amplitude spectra $\{\hat{\boldsymbol{\gamma}}_p\}_{p=1}^P$ are thus obtained as the intermediate dataset for DU-CNN. Then, according to the theoretical clutter covariance matrix, the high-resolution clutter space-time power spectra $\{\mathbf{Z}_{T,p}\}_{p=1}^P$ are obtained based on (17), which is used as the output label dataset for DU-CNN.

Step 3. Via the previous two steps, the datasets $\{\mathbf{x}_p\}_{p=1}^P$, $\{\hat{\boldsymbol{\gamma}}_p\}_{p=1}^P$ and $\{\mathbf{Z}_{T,p}\}_{p=1}^P$ are obtained. The input dataset and output label dataset of DU-CNN are set as $\{\mathbf{x}_p, \mathbf{Z}_{T,p}\}_{p=1}^P$, the input dataset and output label dataset of ADMM-Net are set as $\{\mathbf{x}_p, \hat{\boldsymbol{\gamma}}_p\}_{p=1}^P$, and the input dataset and output label dataset of the CNN are set as $\{\hat{\boldsymbol{\gamma}}_p, \mathbf{Z}_{T,p}\}_{p=1}^P$. As shown in Figure 5, each of these three datasets is divided into a training dataset with the size as $P_{\text{train}}$ and a testing dataset with the size as $P_{\text{test}}$. The training dataset is used to train the network parameters, whereas the testing dataset is not involved in the training procedure but is used to verify the performance of the trained network.

*3.3. Training Method*

To avoid falling into the local optimum and improving the convergence performance of the network, this paper adopts the method of "pre-training + fine-tuning" to train DU-CNN. First, ADMM-Net and CNN are independently pre-trained based on $\{\mathbf{x}_p, \hat{\boldsymbol{\gamma}}_p\}_{p=1}^P$ and $\{\hat{\boldsymbol{\gamma}}_p, \mathbf{Z}_{T,p}\}_{p=1}^P$, respectively. Then, DU-CNN is trained end-to-end based on $\{\mathbf{x}_p, \mathbf{Z}_{T,p}\}_{p=1}^P$, i.e., the network parameters of ADMM-Net and CNN are fine-tuned jointly. The specific steps can be summarized as follows.

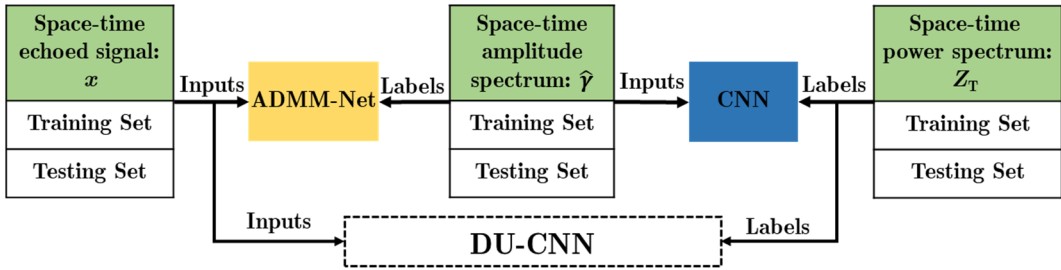

**Figure 5.** Datasets for ADMM-Net, CNN and DU-CNN.

Step 1. According to the iteration parameter setting of the ADMM algorithm, the parameters of ADMM-Net are initialized as $\Theta_A = \{\rho_k = \rho_0, \beta_k = \beta_0, \tau_k = \beta_0\}_{k=1}^{K}$. The network loss function is defined as the mean square error (MSE) between the ADMM-Net output and the label data, which can be expressed as

$$\mathcal{L}(\Theta_A) = \frac{1}{P_{\text{train}}} \sum_{p=1}^{P_{\text{train}}} \left\| \hat{\gamma}^{(K)}(\mathbf{x}_p; \Theta_A) - \hat{\mathbf{Y}}_p \right\|_2^2 \tag{30}$$

where $\hat{\gamma}^{(K)}(\mathbf{x}_p; \Theta_A)$ denotes the output of the $K$-th layer of ADMM-Net with $\mathbf{x}_p$ as the input and $\Theta_A$ as the parameter. The optimal parameter $\Theta_A^* = \{\rho_k^*, \beta_k^*, \tau_k^*\}_{k=1}^{K}$ of ADMM-Net can be obtained by minimizing the loss function through the back propagation method [29], expressed as

$$\Theta_A^* = \arg\min_{\Theta_A} \mathcal{L}(\Theta_A) \tag{31}$$

Step 2. The Glorot method [30] is used to initialize the network parameters of the CNN. Similarly, the network loss function is defined as the MSE of the CNN output and the label data, expressed as

$$\mathcal{L}(\Theta_C) = \frac{1}{P_{\text{train}}} \sum_{p=1}^{P_{\text{train}}} \left\| P_S \hat{\mathbf{Z}}_{C,N}(\hat{\mathbf{Y}}_{S;N,p}; \Theta_C) - \mathbf{Z}_T^p \right\|_F^2 \tag{32}$$

where $\hat{\mathbf{Z}}_{C,N}(\hat{\mathbf{Y}}_{S;N,p}; \Theta_C)$ denotes the output of the CNN with $\hat{\mathbf{Y}}_{S;N,p} = \mathcal{T}(|\hat{\gamma}|_2)/P_S$ as the input and $\Theta_C$ as the parameter. Similarly, the optimal parameters of the CNN can be obtained through the back propagation method by solving the following problem:

$$\Theta_C^* = \arg\min_{\Theta_C} \mathcal{L}(\Theta_C) \tag{33}$$

Step 3. Based on the independent pre-training results of ADMM-Net and CNN, DU-CNN is trained end-to-end to further improve its convergence performance. Here, the parameters of DU-CNN are initialized as $\Theta = \{\Theta_A^*, \Theta_C^*\}$ to avoid the local convergence problem that may occur when directly training DU-CNN. The network loss function is defined as

$$\mathcal{L}(\Theta) = \frac{1}{P_{\text{train}}} \sum_{p=1}^{P_{\text{train}}} \left\| \hat{\mathbf{Z}}_{DC}(\mathbf{x}_p; \Theta) - \mathbf{Z}_{T,p} \right\|_F^2 \tag{34}$$

where $\hat{\mathbf{Z}}_{DC}(\mathbf{x}_p; \Theta)$ denotes the output of DU-CNN with $\mathbf{x}_p$ as the input and $\Theta = \{\Theta_A, \Theta_C\}$ as the parameter. Similarly, the optimal parameters of DU-CNN $\Theta^* = \{\rho_k^*, \beta_k^*, \tau_k^*\}_{k=1}^{K} \cup \{\mathbf{W}_e^*, \mathbf{b}_e^*\}_{e=1}^{E}$ can be obtained as follows:

$$\Theta^* = \arg\min_{\Theta} \mathcal{L}(\Theta) \tag{35}$$

## 4. Simulation Results

In this section, the performance of the DU-CNN-STAP method is verified via simulations. A comparative analysis with the CNN-STAP method and the SR-STAP method is also performed. Considering the computational complexity, dataset size and memory usage, the parameters shown in Table 1 are used for simulations to mimic an airborne radar in a clutter environment [14]. For the SR-STAP algorithm, the iteration parameters of the ADMM algorithm are set as $\rho_0 = 1.0$, $\beta_0 = 0.01$, $\tau_0 = 0.04$ and $K_0 = 2000$. For the proposed method, the number of network layers of the CNN in DU-CNN is $E = 5$, and the convolutional dimensions $\{c_e \times f_e \times f_e \times n_e\}_{e=1}^E$ are set as $(1 \times 11 \times 11 \times 16)$, $(16 \times 9 \times 9 \times 8)$, $(8 \times 7 \times 7 \times 4)$, $(4 \times 5 \times 5 \times 2)$ and $(2 \times 3 \times 3 \times 1)$. For the CNN-STAP method, the CNN is set as the same as that in the proposed method. In the training process of ADMM-Net, the CNN and DU-CNN, the batch size is set as 128. All network parameters are optimized via the Adam optimizer, and the learning rates for different networks are set as $10^{-4}$, $5 \times 10^{-3}$ and $2 \times 10^{-5}$, respectively.

**Table 1.** Simulation parameters.

| Parameter | Symbol | Value |
|---|---|---|
| Airplane height | $H$ | 3000 m |
| Element number in ULA | $M$ | 10 |
| Pulse number in CPI | $N$ | 10 |
| Element spacing | $d$ | 0.3 m |
| Signal wavelength | $\lambda$ | 0.6 m |
| Elevation angle | $\varphi$ | $U(5°, 45°)$ |
| Non-side-looking angle | $\theta_e$ | $U(-30°, 30°)$ |
| Clutter ridge slope | $\beta$ | $U(0.2, 5)$ |
| Clutter-to-noise ratio | $CNR$ | $U(30, 50)$ dB |
| Number of clutter blocks | $N_c$ | 181 |
| Spatial frequency range | $[f_{s,min}, f_{s,max}]$ | $[-0.5, 0.5]$ |
| Doppler frequency range | $[f_{d,min}, f_{d,max}]$ | $[-0.5, 0.5]$ |
| Number of spatial frequencies | $N_s$ | 50 |
| Number of Doppler frequencies | $N_d$ | 50 |
| Number of training data | $P_{train}$ | 10,000 |
| Number of testing data | $P_{test}$ | 2000 |

### 4.1. Network Convergence Analysis

In this sub-section, the training convergences of ADMM-Net, the CNN and DU-CNN are analyzed. A comparison with the training convergence of the CNN in the CNN-STAP method [19] is also provided. The training convergences of ADMM-Net with the number of network layers $K$ as 20, 30 and 40 are given in Figure 6a. It can be seen that the loss function of ADMM-Net decreases gradually and remains unchanged after about 150 epochs. As the number of network layers increases, the loss of ADMM-Net decreases, and its convergence performance improves. Considering the computing complexity and the convergence performance, the number of network layers of ADMM-Net is set as $K = 20$ in the following simulations. Figure 6b provides a comparison of the convergences of CNNs between the proposed method and the CNN-STAP method. It can be seen that, under the condition with the same CNN network scale, because the input of the CNN in the proposed method is the high-resolution clutter space-time spectrum obtained by the ADMM algorithm, it has a higher performance than that of the CNN-STAP method with the Fourier-transform-based (i.e., DBF-based) low-resolution clutter space-time spectrum as its input. Figure 6c shows the convergences of the proposed DU-CNN and the CNNs in the CNN-STAP method, where the number of network layers in CNN-New is 7 and the convolutional dimensions are $(1 \times 11 \times 11 \times 16)$, $(16 \times 9 \times 9 \times 12)$, $(12 \times 9 \times 9 \times 10)$, $(10 \times 7 \times 7 \times 8)$, $(8 \times 5 \times 5 \times 4)$, $(4 \times 5 \times 5 \times 2)$ and $(2 \times 3 \times 3 \times 1)$. It can be seen that (1) DU-CNN has a higher convergence performance than the CNN in the CNN-STAP

method with the same parameters, and (2) increasing the network scale can improve the convergence performance of the CNN in the CNN-STAP method, obtaining the result close to DU-CNN, but its computing complexity is increased.

In general, as the network layer number and scale become larger, the nonlinear transform capability of DU-CNN becomes stronger, but the computing burden is increased. In practice, to determine the appropriate network layer number and scale of DU-CNN under different conditions, the following approach can be used: (1) conduct off-line training of DU-CNN with different network layer numbers and scales (make sure the network training converges), (2) obtain the condition for which the increase in the layer number and scale does not decrease the training loss significantly and (3) considering the balance of the clutter suppression performance and computational complexity, choose the network layer number and scale under the above-mentioned condition for DU-CNN.

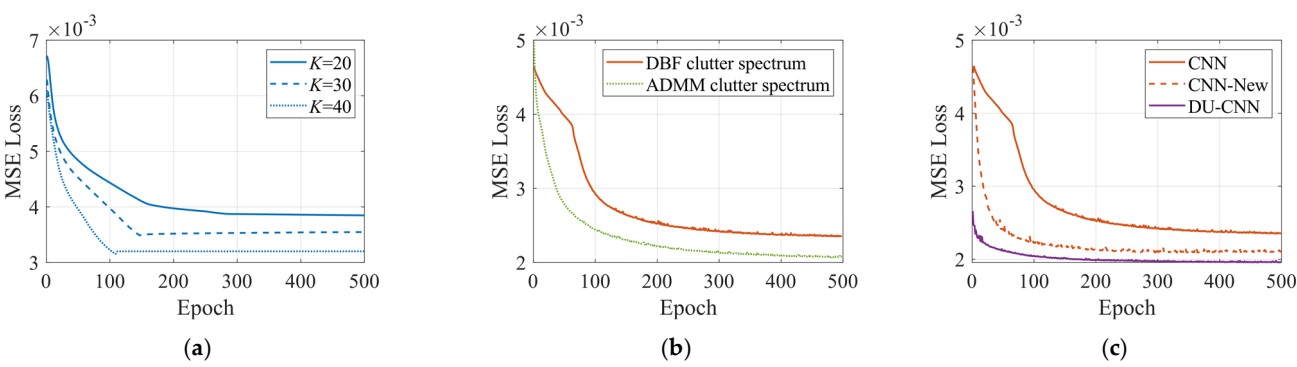

**Figure 6.** Training convergences of different networks. (**a**) ADMM-Nets with different layer numbers. (**b**) CNNs with different inputs. (**c**) Different STAP networks.

### 4.2. Clutter Suppression Performance

In this sub-section, the clutter suppression performance of DU-CNN-STAP is verified and compared with the CNN-STAP and SR-STAP methods. For convenience, the SR-STAP method still adopts the ADMM algorithm; hence, it is also named ADMM-SATP. The clutter space-time spectra estimated by different methods are shown in Figures 7 and 8, where Figure 7 corresponds to the side-looking case with the simulation parameters as $\varphi = 29.6°$, $\beta = 1$, $\theta_e = 0°$ and CNR = 42.8 dB and Figure 8 corresponds to the non-side-looking case with the simulation parameters as $\varphi = 26.5°$, $\beta = 1.7$, $\theta_e = -5.7°$ and CNR = 49.4 dB.

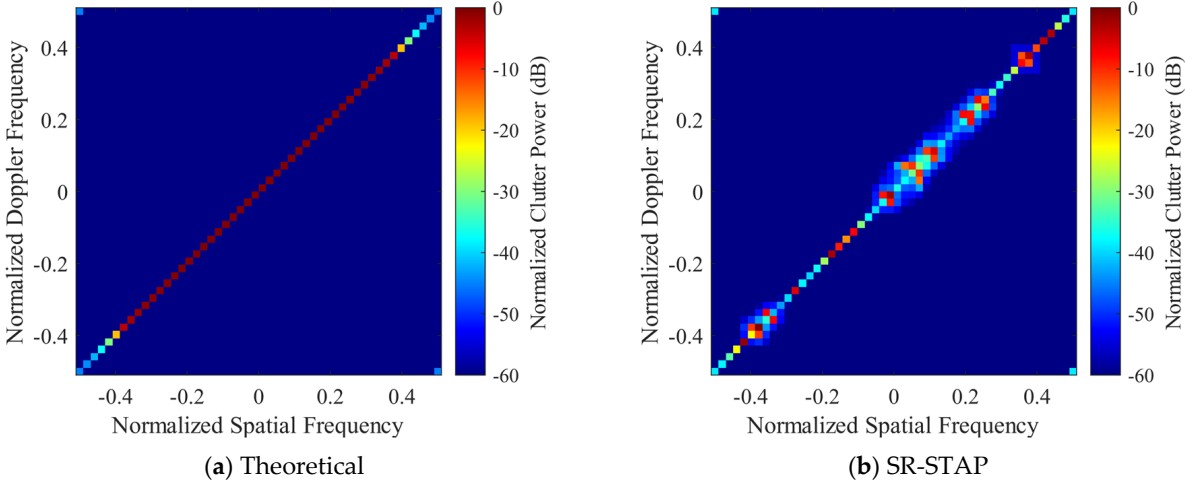

**Figure 7.** *Cont.*

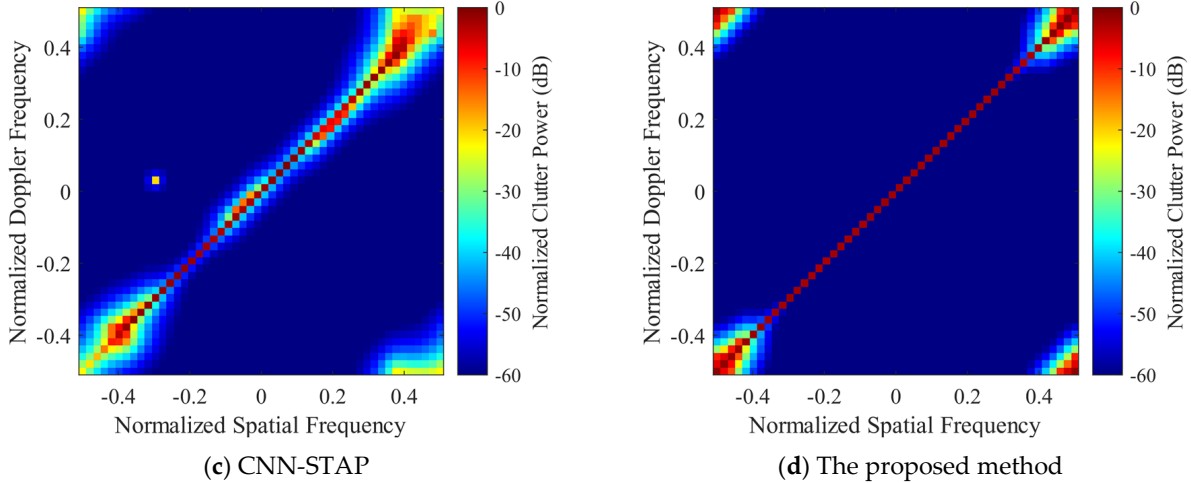

**Figure 7.** Clutter spectrum estimation results with $\varphi = 29.6°$, $\beta = 1$, $\theta_e = 0°$ and CNR = 42.8 dB.

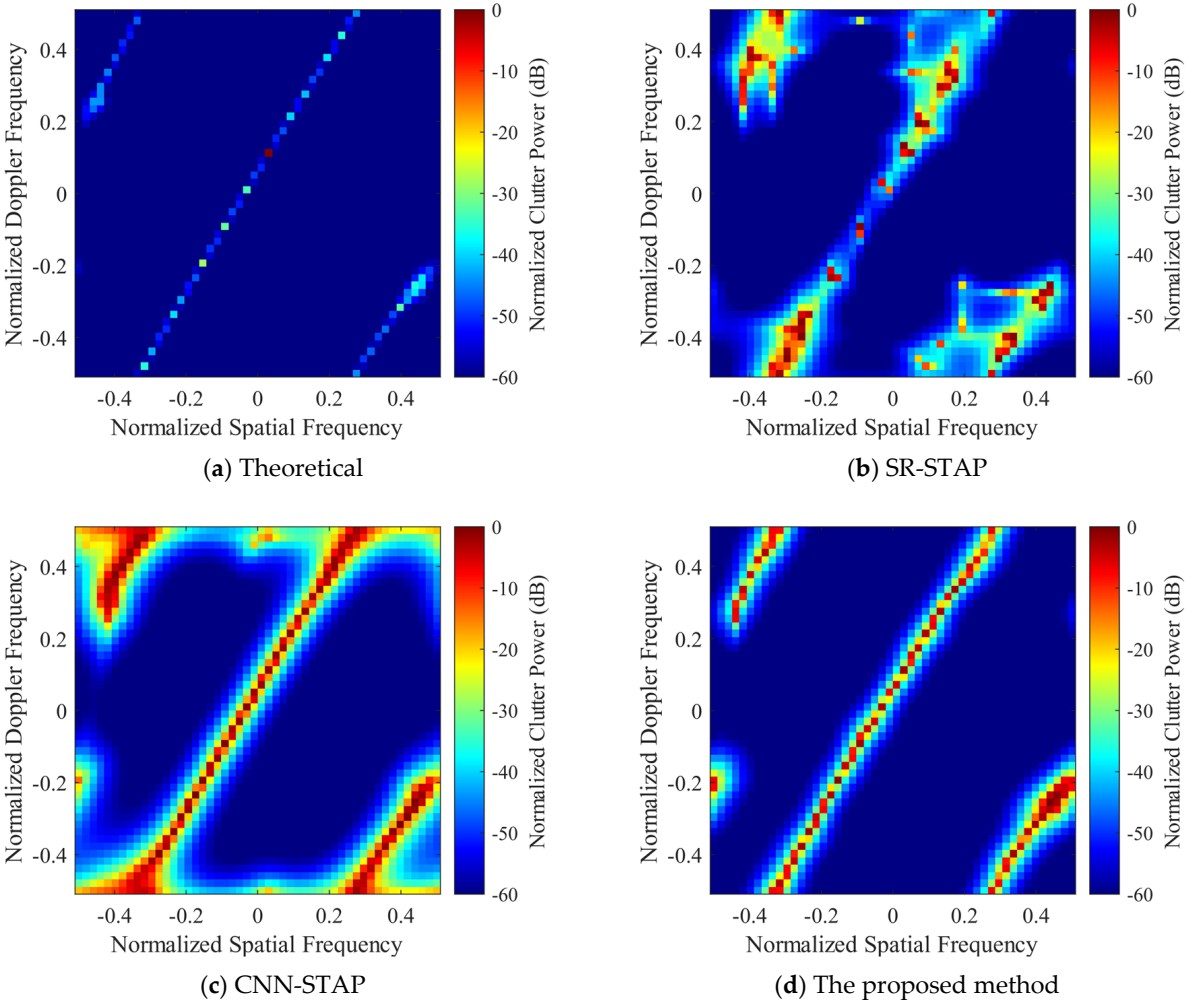

**Figure 8.** Clutter spectrum estimation results with $\varphi = 26.5°$, $\beta = 1.7$, $\theta_e = -5.7°$ and CNR = 49.4 dB.

It can be seen that the SR-STAP method can obtain a high clutter space-time spectrum estimation performance under the side-looking condition; however, an uneven clutter distribution problem occurs. As the clutter sparsity becomes worse under the non-side-looking case, there are some interferences deviating from the clutter ridge, and the performance of the SR-STAP method deteriorates severely. Under the two conditions, the CNN-STAP

method can reconstruct the clutter space-time spectrum effectively. The clutter distribution is continuous, and there is less interference deviating from the clutter ridge. However, this method broadens the clutter ridge. The proposed DU-CNN-STAP method can obtain a higher performance compared to those of the CNN-STAP and SR-STAP methods, with the obtained clutter space-time spectrum estimation results being close to the theoretical ones.

Based on the clutter space-time spectra shown in Figures 7 and 8, the clutter suppression performance of different methods is shown in Figure 9 by using the SCNR loss as the indicator. The spatial frequency of the target is set to 0, and its normalized Doppler frequency varies linearly in the range of $[-0.5, 0.5]$. It can be seen that the SR-STAP and CNN-STAP methods can generate a deep null close to the zero frequency under the side-looking condition, providing good suppression of the clutter. However, unevenly distributed clutter, interferences deviating from the clutter ridge and the broadened clutter ridge make the null formed by the SR-STAP method wider, reducing its slow target detection performance. In contrast, the DU-CNN-STAP method proposed in this paper can benefit from a higher clutter spectrum estimation performance; thus, the width and depth of the null are well controlled to obtain a high clutter suppression performance.

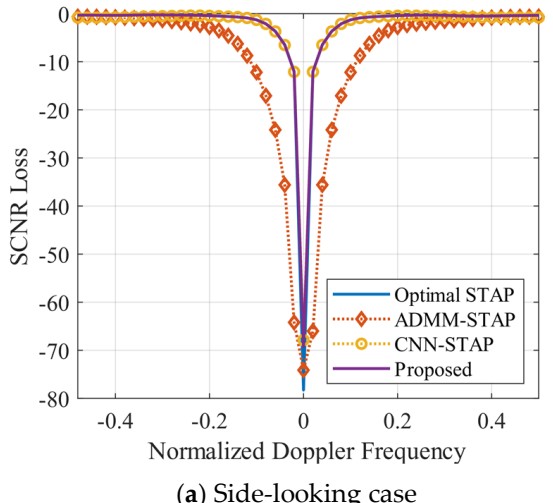

**(a)** Side-looking case

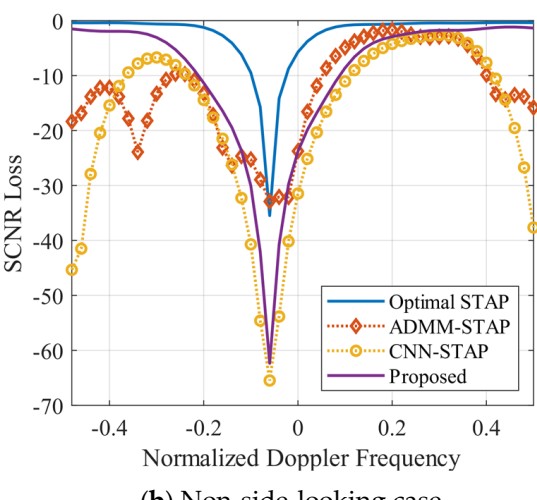

**(b)** Non-side-looking case

**Figure 9.** SCNR loss curves of different methods.

### 4.3. Computational Complexity Analysis

In this sub-section, the computational complexity of the proposed method is analyzed, and a comparison with the SR-STAP and CNN-STAP methods is also provided. It should be emphasized that, because the method of offline training and online application can be used, the computational complexity analysis in this paper does not consider the computational complexity required for network training. Using the number of multiplications as the indicator, the computational complexity of the Fourier-transform-based spectrum estimation method in (16), the ADMM algorithm in (25) and the CNN method in (27–29) are shown in Table 2. It should be noted that the trained ADMM-Net has the same operations as those of the ADMM algorithm. Thus, under the same iteration number (network layer), the computational complexities of ADMM-Net and ADMM are the same. According to Table 2, the computational complexities of the proposed method, the SR-STAP method and the CNN-STAP method for clutter space-time spectrum estimation are $C_A(K) + C_C$, $C_A(K_0)$ and $C_F + C_C$, respectively. The resulting computational complexities of different methods under the condition of $M = N = N_d/5 = N_s/5 = 4 \sim 16$ are shown in Figure 10, where CNN-STAP-New corresponds to CNN-New in Figure 6c, whose network scale is increased to improve the performance of CNN-STAP. It can be seen that the computational complexities of the proposed method and the CNN-STAP method are much lower than that of the SR-STAP (i.e., ADMM-STAP) method. As the dimensionality of the estimation problem becomes higher, the advantage becomes more obvious. Under the condition that

the number of array elements is larger, the computational complexity of the proposed method is higher than that of the CNN-STAP method. However, to obtain a performance similar to that of the proposed method, the computational complexity of the CNN-STAP method increases.

**Table 2.** Computational complexities of different methods.

| Method | Symbol | Number of Multiplications |
|---|---|---|
| Fourier | $C_F$ | $(MN + 1)N_d N_s$ |
| ADMM | $C_A(K)$ | $2MNN_d N_s K$ |
| CNN | $C_C$ | $\sum_{e=1}^{E} c_e f_e^2 n_e N_d N_s$ |

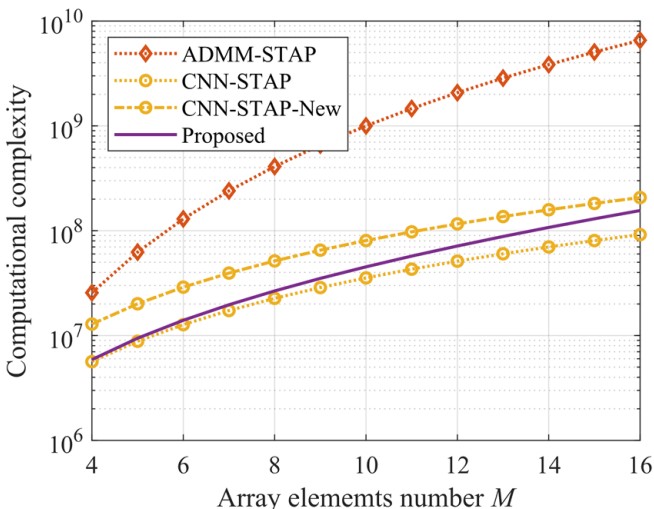

**Figure 10.** Computing complexities of different methods under different conditions.

## 5. Measured Data Processing Results

In this section, the practical performance of the proposed DU-CNN-STAP method is verified via the Mountain-Top actual measured data [14]. A comparison with the SMI-STAP method, the SR-STAP method and the CNN-STAP method is also provided, where the parameter settings for SR-STAP, CNN-STAP and the proposed DU-CNN-STAP method are the same as those in Section 4. In the Mountain-Top measured data, the number of array elements and pulses is 14 and 16, and the target with a normalized Doppler frequency as 0.25 is located at the 147th range cell. In order to match the simulations, the data corresponding to 10 elements and 10 pulses are taken for processing. It is assumed that there is no array element error, and the number of guard range cells is set as 12. The SMI-STAP method takes 200 training range cells around the 147th range cell for estimation, and the other STAP methods take the data of the 154-th range cell for estimation.

Obtained via different STAP methods, the clutter space-time spectrum estimation results are shown in Figure 11, and the space-time filters are then designed for clutter suppression, giving the target detection results shown in Figure 12. It can be seen that the DU-CNN-STAP method proposed in this paper can obtain better results in processing the measured data. Its clutter space-time spectrum estimation is closest to the result of the SMI-STAP method, and its target detection performance is higher than that of the other two STAP methods.

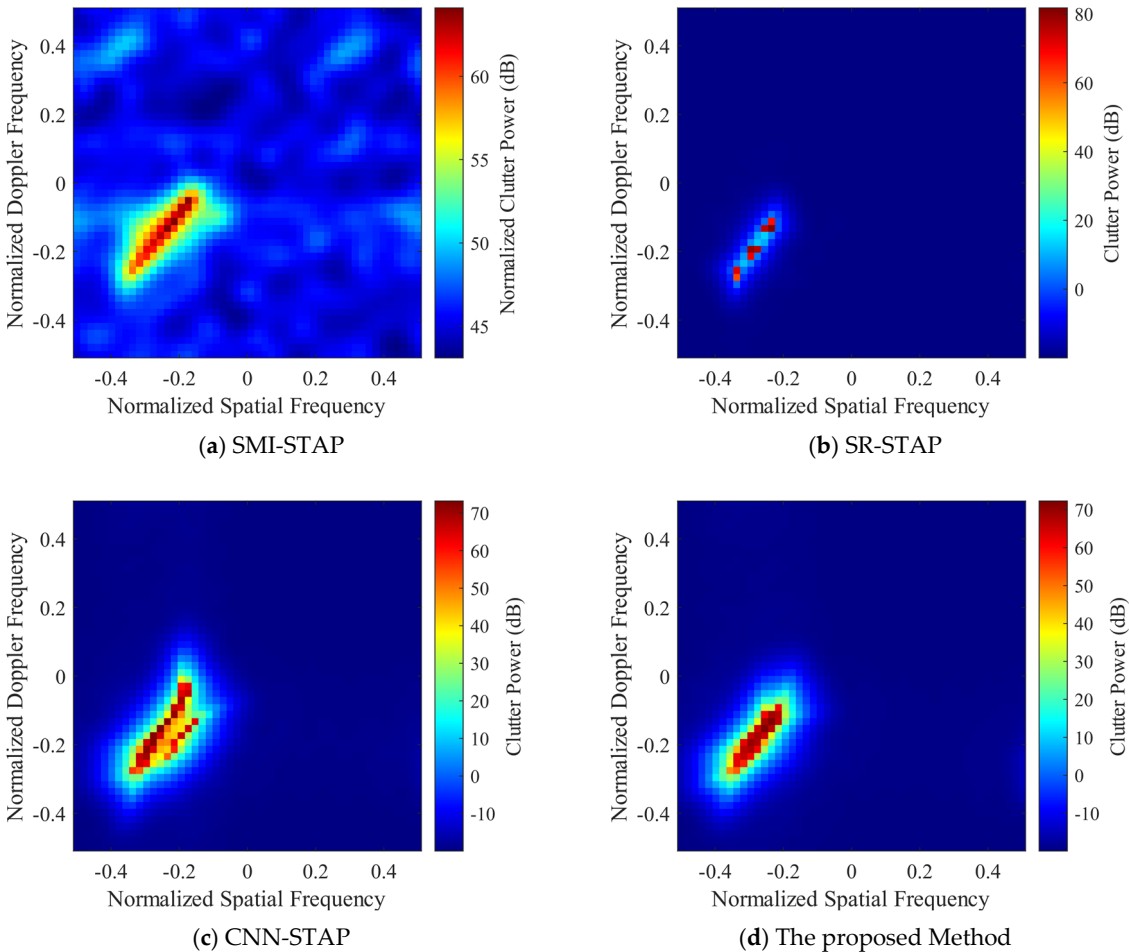

**Figure 11.** Clutter spectrum estimation results of Mountain-Top actual measured data.

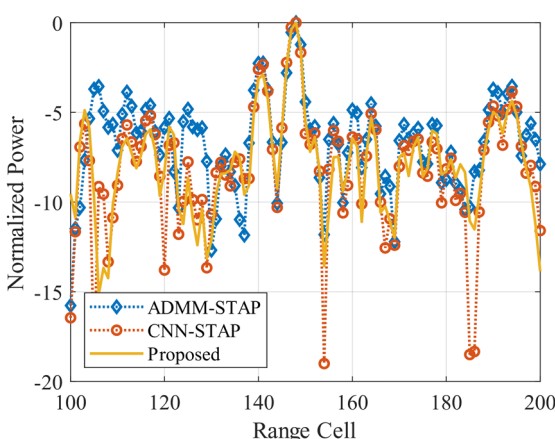

**Figure 12.** Target detection results of Mountain-Top actual measured data.

## 6. Conclusions

In this study, the DU-CNN-STAP method is proposed for airborne radar clutter suppression and target detection, and its processing framework, network structure, dataset construction and training methods are described in detail. Simulation and experimental results under different conditions show that, compared with the SR-STAP method, the proposed method can improve the clutter space-time spectrum estimation performance and reduce the computational complexity. Compared to the CNN-STAP method, the proposed method can reduce the requirement for network reconstruction capability and obtain

higher clutter suppression performance. Future research will focus on the performance improvement of the proposed method under non-ideal conditions (e.g., training range cells are contaminated by moving targets), the construction of multi-dimensional deep unfolding networks and the optimization of CNN network structures.

**Author Contributions:** Conceptualization, B.Z. and W.F.; methodology, B.Z. and W.F.; software, B.Z. and W.F.; validation, B.Z. and H.Z.; writing—original draft preparation, B.Z.; writing—review and editing, W.F. and H.Z.; funding acquisition, W.F. All authors have read and agreed to the published version of the manuscript.

**Funding:** This work was supported in part by the National Natural Science Foundation of China, No. 62001507; Young Talent fund of University Association for Science and Technology in Shaanxi, China, No. 20210106; China Postdoctoral Science Foundation, No. 2021MD703951; and Youth Talent Lifting Project of the China Association for Science and Technology, No. 2021-JCJQ-QT-018.

**Data Availability Statement:** The data presented in this study are available on request from the corresponding author.

**Conflicts of Interest:** The authors declare no conflict of interest.

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
