# Peer review of "Airborne Radar STAP Method Based on Deep Unfolding and Convolutional Neural Networks"

_electronics, doi:10.3390/electronics12143140_

Round 1

Reviewer 1 Report

the paper is well written and does a good job covering a variety of practical simulated cases, comparing the proposed method with other state of the art methods. 

Sufficient detail is provided to describe the algorithm and overall framework, but there are some questions to the authors.

1- How is system (24) solved?

2- How does the algorithm perform if the clutter samples are contaminated by moving target?

3- In Table 1, what happens if the simulation parameters increase? kindly, briefly describe the algorithm of selecting those values

Author Response

Dear Reviewer,

Thank you very much for giving us an opportunity to revise the manuscript entitled “Research on airborne radar STAP method based on deep un-folding and convolution neural network” (ID: electronics-2455187). We express our great appreciation to you and the reviewers for the constructive comments and suggestions.

We have responded to all comments and made corresponding modifications to the manuscript. Revised parts are marked in the revised manuscript. The main corrections and the responses to the comments are appended below.

Thank you again for your work on our manuscript.

Yours sincerely,

Bo Zou and Weike Feng

Reviewer 2 Report

[1] Grammar and writing style need to be improved, professional editing service is recommended.

[2] Line 216, 217 and Figure 2: In the statement “the p th low-resolution clutter space-time power spectrum

input and the p th high-resolution clutter space-time power spectrum output label,” it is not clear how to acquire high-resolution sepctrum from low-resolution spectrum.

It sounds like additional information is generated in some process, please elaborate how this can be done.

[3] Figure 5: Please elaborate the difference between the low-resolution spectrum used for test, as claimed in Section 2, and the high-resolution spectra for training.

[4] Eqn.(27) is difficult to understand, please rephrase.

[5] Lines 458-461: Please elaborate the practical method to determine the range of network layers and scales based on different simulation conditions.

Grammar and writing style need to be improved, professional editing service is recommended.

Author Response

(The authors gave the same response as above.)

Round 2

Reviewer 2 Report

Previous comments have been addressed.

Grammar and writing style can be further improved.

Author Response

Dear reviewer,

Thank you very much for your previous comments and suggestions.

We have checked and modified the manuscript carefully to improve its English writing quality.

The modified parts are marked as yellow in the revised manuscript.

Best regards,

Weike Feng